# Consumption of Bottled Water and Chronic Diseases: A Nationwide Cross-Sectional Study

**DOI:** 10.3390/ijerph21081074

**Published:** 2024-08-15

**Authors:** Jacopo Dolcini, Manuela Chiavarini, Giorgio Firmani, Elisa Ponzio, Marcello Mario D’Errico, Pamela Barbadoro

**Affiliations:** Department of Biomedical Sciences and Public Health, Section of Hygiene, Preventive Medicine and Public Health, Polytechnic University of the Marche Region, 60126 Ancona, Italy; j.dolcini@staff.univpm.it (J.D.); e.ponzio@staff.univpm.it (E.P.); m.m.derrico@staff.univpm.it (M.M.D.); p.barbadoro@staff.univpm.it (P.B.)

**Keywords:** microplastic, plastic pollution, health risk, chronic disease

## Abstract

Plastic pollution is a growing concern. It can form smaller particles called microplastics (<5 mm). Microplastics can break down into even smaller pieces called nanoplastics (<1 μm). These minute particles can infiltrate human cells and tissues, with their health impacts still largely undetermined. On average, a liter of bottled water includes about 240,000 tiny pieces of plastic. The purpose of this study was to evaluate the association between the use of bottled plastic water (BW) and several health outcomes. Utilizing data from the Italian National Institute of Statistics’ “Aspects of Daily Life” survey (N = 45,597), we employed logistic regression to explore the correlation between BW consumption and the prevalence of various chronic diseases, including hypertension, gastric/duodenal ulcers, and kidney stones. Adjustments were made for covariates such as education, age, gender, and economic resources. Our analysis indicated a statistically significant association between BW consumption and increased risk of hypertension (Odds ratio [OR] = 1.05, 95% confidence interval [CI] 1.00–1.11), diabetes (OR = 1.09, 95% CI 1.01–1.18), gastric/duodenal ulcers (OR = 1.21, 95% CI 1.07–1.38), and kidney stones (OR = 1.17, 95% CI 1.03–1.32). The consumption of BW is associated with heightened risk for certain health conditions. Policymakers and healthcare providers should consider implementing targeted prevention strategies and awareness campaigns.

## 1. Introduction

Recent studies have shown how continuous exposure to microplastics may be associated with contaminated sources coming from edible products, including bottled plastic water (BW) [1,2]. When plastics break down over time, they can form smaller particles called microplastics and nanoplastics; a liter of bottled water can include about 240,000 tiny pieces of plastic [3]. These particles, collectively called micro- and nanoplastics (MNPs), are small enough to enter the body’s cells, tissues, and major organs and then interfere with physiological molecular and biological processes, such as protein structure [4] and cell membranes [5].

As a result, this contamination may lead to adverse health effects through an increase in several pathways, such as oxidative stress, inflammation, immune dysfunction, altered biochemical and energy metabolism, impaired cell proliferation, disrupted microbial metabolic pathways, abnormal organ development, and carcinogenicity [6]. In fact, the potential health impacts of MNPs have been a growing area of concern. Recent research suggests that these particles can exacerbate a range of chronic diseases, including hypertension, diabetes, kidney stones, and gastric or duodenal ulcers. Several investigations identify the presence of MNPs in many different organs and systems. It has been clearly shown that MNPs can accumulate in the respiratory system [7], and in vitro studies have revealed how these particles may cause cell damage, inflammation, and apoptosis, triggering the upregulation of pro-inflammatory cytokines and pro-apoptotic proteins like TNF- α, which are associated with airway inflammation in asthma [8]. Evidence from animal studies shows that MNPs can easily accumulate in the gastrointestinal system [9,10], causing a certain amount of cellular damage [11], but their effects on human gut microbiota, inflammation, and associated mechanisms are not yet fully understood. Recent studies have identified microplastics in atheroma plaques [12], suggesting a potential risk for cardiovascular diseases. Furthermore, MNPs may also enter the bloodstream causing adverse effects on the cardiovascular system. Their presence in the human heart tissues has been testified by studies investigating patients undergoing cardiac surgery [13]. Again, MNPs may also play a role in the development of metabolic disorders such as diabetes. Emerging evidence indicates that these particles can disrupt glucose metabolism and insulin signaling pathways. Recent studies have shown that microplastics can cause insulin resistance and altered glucose homeostasis [14]. MNPs have been found in human urine, suggesting that they can pass through the gastrointestinal system into the bloodstream and then be excreted through biological processes [15]. The presence of microplastics in the urinary system raises concerns about their impact on kidney function. Studies have demonstrated that microplastics can accumulate in the kidneys, leading to cellular damage and inflammation. Additionally, cell cultures and animal models have identified the potentially harmful effect of MNPs on kidneys [16,17]. In fact, despite the limited evidence available in the literature, histological and functional alterations have been demonstrated in the kidneys of animal models, as well as cytotoxicity through apoptosis, autophagy, oxidative stress, and inflammation in kidney cells [18]. In summary, the growing body of evidence suggests that MNPs present in bottled water and other consumer products could be linked to a range of chronic diseases, including hypertension, diabetes, kidney stones, and gastric or duodenal ulcers. This study aims to investigate, in a nationwide sample, the possible association between bottled water consumption and the prevalence of these chronic diseases. Specifically, we hypothesize that higher consumption of bottled water is associated with an increased risk of developing these chronic conditions.

## 2. Materials and Methods

### 2.1. Design, Data Source, and Participants

Data on BW consumption were obtained from the “Aspects of Daily Life” survey on households, conducted by the Italian National Institute of Statistics (ISTAT) [19]. The survey is administered each year in a representative sample of the Italian population as part of the integrated system of multipurpose surveys on families. The aim of the survey is to identify a variety of behavioral dimensions and aspects of daily life. The questionnaire is standardized, and the survey is carried out by a sequential CAWI/PAPI mixed-mode technique [19] and has been used for investigations in several studies [20,21,22]. We analyzed data from the 2021 edition of the survey, which included 45,597 individuals and 20,000 families, focusing on those who were 18 years or older at the time of the survey. In this specific instance, the ISTAT derived the populations for the current survey from a compilation of municipalities, which were classified into two groups: a distinct stratum of municipalities with larger populations, labeled as self-representative (SR), and other municipalities, designated as non-self-representative (NSR), which were grouped into strata of similar size based on demographic features. From these strata, two municipalities per stratum were selected with probabilities proportional to their population sizes. For each municipality participating in the survey (both SR and NSR), cluster sampling was utilized. The clusters, consisting of families, were randomly selected from the registry list, and all members of the chosen families were included in the survey. Each municipality had a minimum sample size of 24 families. These families were drawn from the theoretical sample selected for the master sample, and information on the surveyed characteristics of all family members was collected. Nationally, the theoretical sample size of families, determined primarily by cost and operational considerations, was approximately 24,000. To ensure effective control and supervision, the number of sampled municipalities involved should not exceed 900. The allocation of the sample of families and municipalities across the various regions was calculated using a balanced criterion designed to guarantee reliable estimates, both at the national level and within each territorial domain [19].

### 2.2. Variables

The following variables were included in the analysis: educational level (graduate/postgraduate degree, high school diploma, middle school diploma, primary school diploma/none), age (18–44 years old, 45–59 years old, 60–74 years old, ≥75 years old), gender (male/female), economical resources in the last 12 months (1 = inadequate, 2 = scarce, 3 = adequate, and 4 = optimal), body mass index (BMI; 1 = underweight (<18.5 kg/m^2^), 2 = normal (18.5–24.99 kg/m^2^), 3 = overweight (25.00–29.99 kg/m^2^), 4 = obese (>30.00 kg/m^2^)), smoking (1 = current, 2 = former, 3 = never), alcohol consumption (1 = current, 2 = former, 3 = never), physical activity (1 = no, 2 = once or more per week, 3 = once or more per month, 4 = less frequently), hypertension (1 = yes, 0 = no), diabetes (1 = yes, 0 = no), presence of kidney stones (1 = yes, 0 = no), presence of gastric or duodenal ulcer (1 = yes, 0 = no).

### 2.3. Statistical Analysis

Bivariate analyses were performed to study the association between BW consumption and relevant variables, using chi-square tests. Logistic regression models were developed to control for confounding variables and assess the factors independently linked with BW consumption (1 if BW consumption is present; 0 if not). To better assess the models, the Hosmer–Lemeshow test was employed to evaluate the suitability of fit. In cases of poor fit, stepwise regression, using AIC and BIC criteria, was utilized to select between models and discriminate between covariates. Regarding health outcomes, we considered the dichotomized presence of the following chronic diseases: hypertension (1 = yes, 0 = no), diabetes (1 = yes, 0 = no), kidney stones (1 = yes, 0 = no), gastric or duodenal ulcer (1 = yes, 0 = no). We performed several multilevel regression logistic analyses for each chronic disease, specifically hypertension, diabetes, the presence of kidney stones, and the presence of gastric/duodenal ulcers. The level of significance was set to 0.05. Analyses were performed with STATA, version 15 (Stata Corp., College Station, TX, USA).

## 3. Results

A total of 22,217 subjects usually drank BW, representing 56.8% of participants (95% confidence interval, CI 56.27–57.25). In terms of prevalence, bivariate analyses (Table 1) highlighted a higher prevalence of BW consumption in people aged 18 to 44 years (N = 6972, 58.4%, *p*-value < 0.05) with no statistically significant difference between males and females. Considering education, people with a middle school diploma showed the highest prevalence of BW consumption (N = 6111, 59.3%, *p*-value < 0.05), as did people with scarce economic resources (N = 6197, 58.6%, *p*-value < 0.05). Regarding BMI, obese people showed a higher prevalence of BW consumption (N = 2881, 59.3%, *p*-value < 0.05). Considering smoking status, current smokers showed the highest prevalence of BW consumption (N = 4235, 57.9%, *p* value > 0.05). Regarding alcohol consumption, people who never drank alcohol showed a higher prevalence of BW consumption (N = 5809, 60.5%, *p* value < 0.05). Considering physical activity, people who were active less frequently showed a higher prevalence (N = 1910, 59.8%, *p* value < 0.05) Focusing on stratification according to health outcomes, higher prevalence was found among people with diabetes (N = 1898, 59.2%, *p*-value ≤ 0.05), gastric/duodenal ulcers (N = 662, 61.6%, *p*-value ≤ 0.05), and kidney stones (N = 700, 60.7%, *p*-value ≤ 0.05). Concerning the presence of hypertension, even if people affected by it showed a higher prevalence (N = 5551, 57.3%, *p*-value > 0.05) at the bivariate analysis, the difference with people not affected did not reach statistical significance. As shown in Table 2, the consumption of BW was associated with the presence of hypertension (OR = 1.05, C.I. 1.00–1.11, *p*-value 0.05), diabetes (OR = 1.09, C.I. 1.01–1.18, *p*-value = 0.005), kidney stones (OR = 1.17, C.I. 1.03–1.32, *p*-value = 0.013), and gastric or duodenal ulcer (OR = 1.21, C.I. 1.07–1.38, *p*-value = 0.003). All models were adjusted for age, sex, level of education, economic resources, and BMI.

## 4. Discussion

The results of this study showed interesting associations between the consumption of BW and several chronic diseases, including hypertension, diabetes, kidney stones, and gastric or duodenal ulcers. This finding has important public health implications, considering that over 56% of our sample reported using BW. Furthermore, our data differ from a survey by CSA Research, which indicates that about 70% of Italians rely on bottled water for their daily hydration needs (ALOR Italy). These findings may be in accordance with previous studies that observed an association between hypertension and patients with high levels of di(2-ethylhexyl) phthalate (DEHP), a synthetic chemical commonly used as a plasticizer additive [23]. Also, two other plasticizer additives, bisphenol A and bisphenol S, have been shown to be associated with an increased risk of type 2 diabetes in a select case-cohort study [24]. Even pre-pathological conditions, such as pre-diabetes, showed a higher likelihood of association with BW consumption [25]. These effects on cardiovascular and endocrine systems are likely to be multifactorial, but both population-based and experimental studies point to inflammation, oxidative stress, and hormone imbalances as potential mediators. Chemical compounds that are usually present in the composition of plastics have been seen to have endocrine-disrupting properties and can alter both hormone homeostasis and signal transduction pathways [26], thereby increasing the risk of adverse health outcomes. So, from cellular and endocrine signal disruption, chronic exposure may lead to relevant and manifest clinical outcomes as found in our study’s population. Regarding kidney disease, again our results seem to be in agreement with what is actually known about plastic exposure and the higher prevalence of kidney health issues. It has been observed that melamine, a versatile compound with many industrial applications, including plastic production, can leak from bottles, plates, cups, and utensils into food and water when exposed to acids or high temperatures. This compound can then cause renal tubular cell injury through inflammation, fibrosis, and apoptosis, suggesting that melamine-induced apoptosis and/or necrosis may subsequently result in acute kidney injury and promote kidney stone formation [27]. Moreover, as well as cardiovascular diseases, a recent systematic review and meta-analysis showed that, also in kidney diseases, bisphenol A has been counted among possible risk factors. An in vitro study showed that MNPs can enter the cells through endocytosis, causing damage to cellular microstructures and an increase in the expression of JNK1/2/3 and TNF-α, a pathway involved in cancer-related pathways [28]. Regarding the gastrointestinal tract (GIT), we found an increased likelihood between the assumption of BW and the presence of gastric or duodenal ulcers. Some animal models have shown MNPs can cause intestinal barrier dysfunction by epithelial cell apoptosis through ROS production [29], and other studies have found that the presence of several types of plastic particles can alter the gut microbial composition, diversity, and metabolic pathways, which may disrupt nutrient metabolism, ultimately breaking gut homeostasis and promoting inflammation [30].

To our knowledge, this is the first study in Italy to investigate the association between consumption of BW and the presence of several chronic diseases, such as hypertension, diabetes, kidney stones, and gastric/duodenal ulcers. Moreover, our data covered a nationwide sample of thousands of subjects, and interviews and data collection were conducted under rigorous methodological methods, since they were carried out by ISTAT-trained personnel.

Nevertheless, some limitations should be acknowledged. Due to the observational nature of this study, we cannot claim any causal link between BW consumption and the presence of the investigated chronic diseases. Moreover, since all data are declarative, there is potential for declarative or recall bias in our sample. Specifically, we did not have information on the exact quantity of BW consumption in terms of bottles, liters, or other quantifiable measures that could have been useful in better estimating the associations with chronic diseases.

We used a proxy based on consumption declarations, but this can generate error estimates because of an imprecise quantification of exposure or, again, recall bias. Another important aspect to highlight is that we did not include dietary habits in our models. Even though we recognized the importance of this confounder, we did not have information on general dietary habits, in terms of the number of macronutrients consumed or nutritional lifestyles such as vegan, vegetarian, or omnivorous diets. Moreover, we did not have information on the possible presence of plastic nanoparticles in different types of food. It could be biased to consider some kinds of foods riskier than others based on previous perspectives. Such perspectives are considered in order to evaluate, for example, the increased risk of several diseases associated with a higher level of consumption of meat rather than fruits and vegetables, because red and cured meat may increase exposure to cholesterol, salt, ammonium nitrites and nitrates, hormones, antibiotics, and many other substances. Adding the consumption of several different foods would introduce possible bias due to these several uncontrolled factors and substances. To address this, we included BMI in our models. Beyond being an important confounder for cardiovascular diseases such as hypertension and diabetes, BMI can be a reasonable proxy for dietary habits. However, we recognize the need for more studies focused on this aspect to evaluate the presence of MPNs through specific techniques aimed at detecting their presence in different kinds of foods. A more precise quantification could offer a better implementation of these covariates in the models aimed at evaluating possible associations between plastic exposure and diseases. In the end, it must be noted that the official language of ISTAT, the government institution conducting the survey, is Italian, and since the survey administration was not conducted in other languages, this could have caused a selection bias.

## 5. Conclusions

Our data raise some important questions regarding public health. Despite several worldwide efforts, the production and use of plastic is constantly increasing [31]; and a growing body of evidence points out that the presence of plastics poses potential risks to human health via ingestion [32]. Among exposure pathways, BW has been found to be a possible major source of exposure due to the higher presence of micro- and nano plastic particles due to possible breakdown over time. This exposure can reach estimated concentration values 10 to 100 times greater than those estimated in the past [3]. These minuscule particles can invade individual cells and tissues in several major organs and systems, interfering with cellular processes and metabolism and depositing endocrine-disrupting chemicals, such as bisphenols, phthalates, flame retardants, per- and polyfluorinated substances (or PFASs), and heavy metals. These multiple exposures can lead to serious adverse health effects, and it is crucial to increase studies and knowledge on these possible effects and how they are associated with plastic exposure. Policymakers and public health institutions can no longer delay initiatives aimed at reducing plastic production, consumption, and use and plastic’s association with water consumption. To obtain these goals, they can adopt several pathways, such as encouraging the use of tap water and other safe, plastic-free water sources, through public health campaigns. They should provide information on the safety and benefits of these alternatives compared with bottled water. Moreover, they may increase efforts to sustain industries in terms of the research and development of safer packaging alternatives that can minimize the release of MNPs into the environment. Additionally, regulatory frameworks may need revision in terms of strengthening already existing standards that need stricter indications for plastic production and disposal. Research efforts need to be increased to update current knowledge around both the ongoing monitoring of plastic contamination and possible pathways and mechanisms for plastic-related health effects that have not yet been fully elucidated. On the other hand, at the citizen and end-user level, information and preventive campaigns must be implemented at both national and international levels to increase the empowerment of citizens and their awareness in terms of choosing healthier and alternative methods of drinkable water consumption. Looking at a wider perspective, this can be extremely important also in the context of a ‘One Health’ approach, since increasing plastic-free water consumption may result in benefits not only for humans but also for animal health and the environment. In this perspective, global partnerships may be recommended to address the issue of plastic pollution and its health impacts. Sharing knowledge, resources, and best practices across borders could be very effective in tackling this problem on a global scale.

## Figures and Tables

**Table 1 ijerph-21-01074-t001:** Distribution of proportion of people drinking BW, according to selected personal characteristics.

		N	% of People Drinking BW	*p*
Sex	Males	10,550	56.6	0.55
	Females	11,667	56.9	
Age class (years)	18–44	6972	58.4	0.00
	45–59	6249	55.8	
	60–74	5587	56.5	
	≥75	3409	55.7	
	NA	468	/	
Education level	Graduate/post-graduate degree	3295	51.4	0.00
	High school diploma	8598	56.5	
	Middle school diploma	6111	59.3	
	Primary school diploma/none	3905	59.2	
	NA	308	/	
Economical resources in last 12 months	Optimal	296	49.7	0.00
	Adequate	14,976	56.1	
	Scarce	6197	58.6	
	Inadequate	749	57.9	
BMI	Normal	10,934	56.1	0.00
	Underweight	591	53.7	
	Overweight	7811	57.1	
	Obese	2881	59.3	
Smoking status	Current	4235	57.9	0.06
	Former	5605	56.1	
	Never	12,377	56.7	
Alcohol consumption	Current	15,279	55.4	0.00
	Former	940	56.2	
	Never	5809	60.5	
	NA	189	/	
Physical activity	No	15,053	57	0.00
	Once or more per week	3564	55	
	Once or more per month	1690	55.2	
	Less frequently	1910	59.8	
Hypertension	No	16,053	56.7	0.26
	Yes	5551	57.3	
	NA	613	/	
Diabetes	No	19,293	56.6	0.01
	Yes	1898	59.2	
	NA	1026	/	
Gastric/duodenal ulcers	No	20,238	56.7	0.00
	Yes	662	61.6	
	NA	1317	/	
Kidney stones	No	20,166	56.7	0.02
	Yes	700	60.7	
	NA	1351	/	

NA, Not Available.

**Table 2 ijerph-21-01074-t002:** Chronic diseases associated with BW consumption at logistic regression analysis. All models have been adjusted for sex, age, economic resources, level of education, BMI, smoking status, alcohol consumption, and physical activity.

Outcome	OR	*p*-Value	C.I.
Hypertension	1.05	0.05	1.00–1.11
Diabetes	1.09	0.005	1.01–1.18
Kidney stones	1.17	0.013	1.03–1.32
Gastric/duodenal ulcers	1.21	0.003	1.07–1.38

## Data Availability

The data undergo anonymization and aggregation by ISTAT and are subsequently made publicly accessible for research purposes at the following link: https://www.istat.it/it/archivio/129956 (access on 12 December 2023).

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
