# Peer review of "Consumption of Bottled Water and Chronic Diseases: A Nationwide Cross-Sectional Study"

_ijerph, 2024, doi:10.3390/ijerph21081074_

Round 1
Reviewer 1 Report
Comments and Suggestions for Authors
The introduction provides a good overview of the potential health risks associated with microplastics. Expand the literature review to include more recent and diverse studies on the health impacts of microplastics. Clearly state the hypothesis regarding the association between bottled water consumption and specific chronic diseases.
Provide a stronger rationale for the selection of covariates in the analysis. Consider additional potential confounding factors that could influence the study results, such as dietary habits, physical activity levels, and other lifestyle factors like smoking and alcohol consumption. it is really important to include those variable if available in the data set you used. you need to provide a solid and more concrete explanation on why not you included those variables.
While some limitations are acknowledged, discuss other potential limitations such as measurement errors in self-reported data and lack of information on the exact quantity of bottled water consumed. Additionally, address the absence of key predictors of chronic diseases such as exercise, diet, and other lifestyle factors in the analysis.
Finally, I would recommend that you provide more concrete recommendations for public health interventions based on the study findings.
Comments on the Quality of English LanguageCheck for proper punctuation, especially in complex sentences, to avoid confusion and enhance the flow of the text. For example, in the sentence, "This compound can then cause renal tubular cell injury through inflammation, fibrosis, and apoptosis suggesting that melamine-induced apoptosis and/or necrosis may subsequently result in acute kidney injury and promote kidney stone formation," consider adding a comma after "apoptosis"
I would recommend that the authors review the grammar and the flow of the sentences before they resubmit.
Author Response
Thank you for your review.
In attachment the file.
Best rgards
Manuela Chiavarini, Giorgio Firmani

Reviewer 2 Report
Comments and Suggestions for Authors
In the manuscript titled “Consumption of Bottled Water and Chronic Diseases: A Nationwide Cross-Sectional Study”, the authors have investigated the correlation between the consumption of bottled water with some chronic diseases. The analysis results indicated that some chronic diseases including hypertension, gastric/duodenal ulcers, and kidney stones showed a positive correlation with bottled water consumption. This finding is very interesting and meaningful.
Some comments and suggestions are as follows:
1. Line 91-95, the unit of BMI needs to be kg/m^2.
2. Line 20-21, The full names of terms are suggested when they are used for the first time, such as, “OR”, and “CI”.
3. Line 111-134, results section and Table 1. The percentage values should keep same significant digital numbers in the whole manuscript. For example, Line 111, is with “56.76%”, but Line 114 is with “58.4%”. This applies to the whole manuscript.
4. The volume/amount of bottled water is one of the key factors involved. If this data were included, that would be much more meaningful.
Author Response
Thank you for your review.
In attachment the file.
Best regards
Manuela Chiavarini, Giorgio Firmani

Round 2
Reviewer 1 Report
Comments and Suggestions for Authors
I see that the authors have incorporated the feedback provided for them and I am fine with the updated manuscript.